# Comparative Analysis of Major Flavonoids among Parts of *Lactuca indica* during Different Growth Periods

**DOI:** 10.3390/molecules26247445

**Published:** 2021-12-08

**Authors:** Junfeng Hao, Yuyu Li, Yushan Jia, Zhijun Wang, Rong Rong, Jian Bao, Muqier Zhao, Zhihui Fu, Gentu Ge

**Affiliations:** 1College of Grassland Resources and Environment, Inner Mongolia Agricultural University, Hohhot 010019, China; hjf7756@126.com (J.H.); lyy2017@emails.imau.edu.cn (Y.L.); jys_nm@sina.com (Y.J.); zhijunwang321@126.com (Z.W.); 2Key Laboratory of Forage Cultivation, Processing and High Efficient Utilization of Ministry of Agriculture and Rural Affairs, Inner Mongolia Agricultural University, Hohhot 010019, China; baojian19940203@163.com (J.B.); zhaomuqier@163.com (M.Z.); 3Key Laboratory of Grassland Resources, Ministry of Education, Inner Mongolia Agricultural University, Hohhot 010019, China; fuxiaobao3357@163.com; 4Hinggan League Forestry and Grassland Workstation, Ulanhot 137499, China; 15661245499@163.com

**Keywords:** *L. indica* L. cv. Mengzao, flavonoids, LC-MS/MS, harvest time, dynamic distribution

## Abstract

*L. indica* L. cv. Mengzao, a medicinal plant of the Ixeris genus, is rich in flavonoids. In order to thoroughly analyze the the distribution and dynamic change of major flavonoids in its various parts from different growth periods, the flavonoids extracted from *L. indica* L. cv. Mengzao were identified and quantitatively analyzed by ultra-high-performance liquid chromatography mass spectrometer (LC-MS/MS). Results indicated that 15 flavonoids were identified from *L. indica* L. cv. Mengzao, and rutin, luteolin, luteolin-7-*O*-glucoside, kaempferol, quercetin, and apigenin are the major flavonoids in *L. indica* L. cv. Mengzao. In general, the total flavonoids’ content in different parts of *L. indica* L. cv. Mengzao followed the order flowers > leaves > stems > roots. Flowers and leaves are the main harvesting parts of *L. indica* L. cv. Mengzao, and the flowering period is the most suitable harvesting period. This study provides valuable information for the development and utilization of *L. indica* L. cv. Mengzao and determined the best part to harvest and the optimal time for harvesting.

## 1. Introduction

*Ixeris polycephala* Cass is a kind of ephemeral or perennial herb that derives from the Ixeris Cass. genus. It is native to Europe and Central Asia, and is now mainly distributed in China, Japan, and Vietnam [1]. It is widely used in folk medicine because of its functions of heat-clearing, detoxification, stopping pain, cooling blood, and anti-inflammation [2]. It was first recorded in Shennong’s Classic of Materia Medica (200–300 A.D., Han Dynasty) that *Ixeris polycephala* was used as a medicine. All parts of *Ixeris polycephala* (Figure 1), including the roots, stems, leaves, and flowers, have been used as traditional Chinese medicines for thousands of years [3]. Recently, phytochemical studies showed that flavonoids, triterpenoids, sesquiterpenoids, organic acids, and phenylpropanoids are the main active components responsible for their beneficial efficacies [2]. These compounds have been shown to have a wide range of biological and pharmacological effects, such as anti-tumor [4], protecting the cardiocerebral vascular system [5], and liver protection [6]. These effects have aroused the interest of more and more people, while phytochemicals from *Ixeris polycephala* and their related potential activities have not been fully explored. At present, the annual production of *Ixeris policeohala* in Inner Mongolia is 270 tons per ha, most of which is used as green and succulent fodder for livestock and herbivorous fish, and utilization of its rich medicinal values is still in its initial stage [7]. Therefore, it is urgent to speed up research on how to make full use of *Ixeris polycephala* as a potential product in the food or pharmaceutical industry. In this regard, a thorough analysis of the main chemical components of *Ixeris polycephala* and its related biological activities is essential.

Over the years, scholars from various countries have carried out a series of studies on the flavonoids of the Ixeris genus. The main flavonoids isolated were luteolin, luteolin-7-*O*-β-d-glucoside, apigenin, Luteolin-7-*O*-β-d-glucopyranoside, lutein-7-*O*-β-d-glucoside, Isorhamnetin-3-*O*-β-d-glucopyranosideand, Luteolin-7-*O*-β-d-glucuron- opyranoside ethyl ester, and 5,7-dihydroxy-4′-methoxy-flavone-7-*O*-rutinoside [2,8,9,10]. However, there have been no previous reports on the bioactive components contained in *L. indica* L. cv. Mengzao, which is in the same genus as the already reported species.

To date, there have been studies on the extraction technology from *Ixeris polycephala Cass* [11,12]. However, there are no published reports identifying its active components or its distribution and dynamic accumulation in various parts of *L. indica* L. cv. Mengzao (e.g., roots, stems, leaves, and flowers). In the present study, on the basis of our team’s previous research [13], the flavonoids of *L. indica* L. cv. Mengzao were identified and the contents of major flavonoids in its roots, stems, leaves, and flowers at different growth stages were determined. The study took the optimum extraction conditions and the total flavonoid extracts of *L. indica* L. cv. Mengzao were identifited according to information on parent ion, molecular formula, secondary mass spectrometry ion, and retention time, combined with the thermo mzcloud online database and thermo mzValut local database. Finally, the sensitive and fast ultra-high-performance liquid chromatography mass spectrometer (UPLC-MS/MS) method was used to simultaneously quantify six flavonoids, including luteolin, rutin, quercetin, luteolin-7-*O*-glucoside, apigenin, and kaempferol in various parts of *L. indica* L. cv. Mengzao for the first time, which involved selectively fragmenting the target peak and detecting all the nucleosides of interest through the multiple reaction monitoring (MRM) scanning mode, without ensuring the absolute baseline chromatographic separation of the selected components. Additionally, our study can provide a theoretical basis for carrying out diversified and deep processing of forage. Our data may provide a reference for quality monitoring of *Ixeris polycephala* products and further development and utilization of its resources.

## 2. Results and Discussion

### 2.1. Qualitative Analysis of Flavonoids in L. indica L. cv. Mengzao

In this study, the whole plant samples of *L. indica* L. cv. Mengzao at the initial flowering stage were used as experimental materials and the UPLC-Q-Exactive-Orbitrap- MS/MS (UPLC-QE-MS/MS) method was used to analyze its compounds. In order to detect flavonoids more comprehensively, we carried out this experiment in positive and negative scanning mode. Using information on parent ion, molecular formula, secondary mass spectrometry ion, and retention time, combined with the thermo mzcloud online database and thermo mzValut local database, a total of 15 flavonoids were identified from *L. indica* L. cv. Mengzao. They include eight flavonoid glycosides, six flavonoid aglycones, and one flavanol. The ion flow diagram and characteristics of 15 flavonoid peaks are shown in Figure 2 and Table 1.

The cleavage of flavonoid glycosides is to first remove the corresponding sugar and generate the corresponding aglycons, and then cleave by removing H, OH. and CO or through the specific Retro-Diels-Alder Reaction (RDA) cleavage or cleavage mode II of flavonoids [14]. This includes compounds **2**, **3**, **4**, **5**, **7**, **8**, **9,** and **10**.

Compounds **2**, **4,** and **5** exhibited the same secondary fragment ion *m*/*z* 287. Compound **2** generated an [M + H]^+^ ion at *m*/*z* 595.16504 (C_27_H_30_O_15_). The fragment ion of *m*/*z* 287.05429 was found in the secondary MS, and this ion was a parent ring of kaempferol. The *m*/*z* 433.10678 was caused by the loss of a neutral fragment of glucoside at the molecular ion *m*/*z* 595.16504. Therefore, compound **2** was identified as kaempferol-3-*O*-rutinoside [15]. Compounds **4** and **5** had the same molecular weight, of 448, and fragment ion *m*/*z* 287 and fragment ion *m*/*z* 153 were also found in the second-level mass spectrum of a fragment ion of parent ion, which was split by RDA and accompanied by H transfer. According to the high-resolution mass spectrometry data, the molecular formulas of compounds **4** and **5** are C_21_H_20_O_11_, indicating that they are isomers. Therefore, compounds **4** and **5** were identified as luteolin-7-*O*-glucoside and kaempferol-7-*O*-glucoside combined with other fragment ions [16,17].

Compound **10** generated an [M − H]^−^ ion at *m*/*z* 447.09149 (C_21_H_18_O_11_). In the secondary mass spectrum, *m*/*z* 271.05927 and 153.01793 were the characteristic ions produced by RDA fragmentation at positions C1–C3 of the A ring of the mother nucleus. By comparison with databases and literature [18], compound **10** was identified as apigenin- 7-*O*-glucuronide.

Compound **3** generated an [M − H]^−^ ion at *m*/*z* 463.08777 (C_21_H_20_O_12_). There was a fragment ion peak of *m*/*z* 301.03519 in the secondary mass spectrum, which was the ion peak of aglycon quercetin. From the molecular weight of 464.0877, it can be inferred that this compound is a quercetin monoglycoside. According to the mass spectrum, the fragmentation of the compound produced fragment ions of *m*/*z* 271, 179, and 151, which can be used as characteristic diagnostic ions to determine the correlation between the aglycon of quercetin-3-*O*-glycoside compound and the glycosylation position. From the mass spectrometry information, it can be inferred that *m*/*z* 463.08777 lost one molecule of glucose to obtain aglycon ion *m*/*z* 301.03519. Therefore, compound **3** was identified as quercetin-3-*O*-glucoside [19].

Compound **7** generated an [M − H]^−^ ion at *m*/*z* 461.07242 (C_21_H_18_O_12_). The fragment ion of *m*/*z* 285.04025 in the MS mass spectrum suggested the loss of a glucuronic acid residue in the molecular ion [M − H]^−^ at *m*/*z* 461.07242. Further dissociation of the glycone *m*/*z* 285.04025 ion yielded a series of fragments, m/z 217.08017, 193.48659, and 113.02330. In comparison with the reference compound and literature [20], compound **7** was confirmed as scutellarin. Similarly, compound **9** also had a fragment ion of *m*/*z* 285. It was caused by the loss of a CH_2_O in the molecular ion [M − H]^−^ at *m*/*z* 447.09302. Therefore, compound **9** was identified as astragalin [21].

However, flavonoid aglycones can be cleaved in three ways directly removing H, OH, or CO by cracking, RDA cracking, and cracking mode II [14,22], which includes compounds **1**, **6**, **11**, **12**, **13a** and **15**.

Compound **1** generated an [M − H]^−^ ion at *m*/*z* 609.14600 (C_27_H_30_O_16_). In the MS/MS spectrum, a series of characteristic ions were *m*/*z* 301.03488, 300.02719, 210.46393, 178.99763, and 151.00256. The fragment ion at *m*/*z* 301.03488 is the characteristic fragment ion formed by a parent ion [M − H]^−^ taking off rhamnose and glucose. Therefore, compound **1** was confirmed as rutin [23].

Compound **6** generated an [M − H]^−^ ion at *m*/*z* 285.04037 (C_15_H_10_O_6_). In the MS/MS spectrum, a series of characteristic ions were *m*/*z* 268.68231, 151.50806, and 133.02843. Fragment ions *m*/*z* 151.50806 and 133.02843 are generated by RDA cleavage, which is the characteristic cleavage mode of flavonoids. However, the fragment ion *m*/*z* 268.68231 is the parent ion [M − H]^−^ (loss of hydroxyl radical). Therefore, compound **6** was confirmed as luteolin [24].

Compound **11** generated an [M + H]^+^ ion at *m*/*z* 287.05423 (C_15_H_10_O_6_). In the MS/MS spectrum, the ions *m*/*z* 184.68994 and 87.57372 were the fragment ions of kaempferol cleaved in mode II. Besides, another fragment ion *m*/*z* 153.01776 was formed by the removal of C_8_H_6_O_2_ accompanied by H transfer during the pyrolysis of kaempferol by RDA. Therefore, compound **11** was confirmed as kaempferol [25].

Compound **12** generated an [M – H]^−^ ion at *m*/*z* 301.03506 (C_15_H_10_O_7_). In the MS/MS spectrum, the fragment ion *m*/*z* 273.04022 and 151.00264 was fromed by the parent ion losing a neutral molecule CO and the parent ion breaking through RDA. Therefore, compound **12** was confirmed as quercetin [26].

Compound **13** generated an [M − H]^−^ ion at *m*/*z* 269.04504 (C_15_H_10_O_5_). In the MS/MS spectrum, the fragment ion *m*/*z* 151.00225 was formed by the removal of C_8_H_6_O during the pyrolysis of the flavonoid mother nucleus by RDA. Then, one molecule of CO_2_ was removed by ion *m*/*z* 151.00225 to obtain fragment ions of *m*/*z* 107.01238. Therefore, compound **13** was confirmed as apigenin [27,28].

Compound **15** generated an [M − H]^−^ ion at *m*/*z* 299.05594 (C_16_H_12_O_6_). In the MS/MS spectrum, the fragment ion *m*/*z* 284.03229 was obtained by the loss of a molecule of CH_3_ from the parent ion *m*/*z* 299.05594. Therefore, compound **15** was confirmed as hispidulin [29].

In addition, we identified three other flavonoids from *L. indica* L. cv. Mengzao. Compounds **8**, **9,** and **14** were identified as quercetin-3-*O*-malonylglucoside, astragalin, and isokaempferide by comparing the retention time and mass spectrometry data with the reference substance, respectively.

By adding internal standards, the concentration of 15 flavonoids in *L. indica* L. cv. Mengzao could be initially obtained. Figure 3 shows that rutin, luteolin, luteolin-7-*O*-glucoside, kaempferol, quercetin, and apigenin are the high concentration in *L. indica* L. cv. Mengzao and reached above 50 μg/mL. Thus, we used them as the major flavonoid components for the next study.

### 2.2. Quantitative Analysis of Major Flavonoids in Different Parts of L. indica L. cv. Mengzao

#### 2.2.1. Optimization of Mass Spectrometry and Chromatographic Conditions

Through preliminary experiments, the best mass spectrometry conditions were obtained. All standard solutions were implanted in positive and negative ion modes by the MS method. The results showed that luteolin, luteolin-7-*O*-glucoside, and apigenin had stronger response in positive ion mode than in negative ion mode. The responses of rutin, quercetin, and kaempferol in negative ion mode were higher than those in positive ion mode. The multi-reaction monitoring parameters of six compounds are shown in Table 2.

It was crucial to optimize the chromatographic conditions, such as the choices of mobile phase and elution program. The chromatographic parameters were optimized to achieve higher chromatographic resolution and reduce analysis time. In the present study, the methanol-water system, acetonitrile-water system, and acetonitrile −0.1% formic acid water system were investigated as mobile phases. It was found that, compared with methanol, acetonitrile was a polar aprotic solvent and produced a narrow peak in a short time. Therefore, acetonitrile was the best organic phase. The results showed that 0.1% formic acid in water/acetonitrile had a good chromatographic separation. The combination was selected for gradient elution. This result is also consistent with previous studies [30]. Comparing the separation effects of isocratic elution and gradient elution on six flavonoids, we found that isocratic elution could not effectively separate the target compounds. Therefore, gradient elution was selected. Finally, the optimum separation conditions for the separation of six flavonoids from *L. indica* L. cv. Mengzao were determined. The mobile phase was 0.1% formic acid in water (A) and acetonitrile (B), and the gradient elution procedure was as follows: 0~2 min, 95% A; 2~4 min, 95~70% A; 4~7 min, 70~60% A; 7~10 min, 60~20% A; 10~13 min, 20~5% A; 13~15 min, 5% A; 15~17 min, 5~50% A; and 17~20 min, 50~95% A. The typical chromatograms of six compounds are shown in Figure 4.

#### 2.2.2. Method Validation

Through linear calibration curves, limit of detection (LOD), limit of quantification (LOQ), precision, repeatability, stability, and recovery of the LC-MS method, the quantitative determination of flavonoids in *L. indica* L. cv. Mengzao was established. The results are shown in Table 3 and Table 4. All calibration curves exhibited an excellent coefficient of determination (R ≥ 0.9990). The RSD values of six standards were less than 4.08%, and the RSD values of stability and repeatability were less than 4.56%. The results showed that the established method was accurate enough and could be used to determine six flavonoids in different parts of *L. indica* L. cv. Mengzao at different growth stages.

#### 2.2.3. Dynamic Characteristics of Six Major Flavonoids in Different Parts of *L. indica* L. cv. Mengzao

The harvest period is an important factor affecting the quality of medicinal materials. The accumulation of secondary metabolites in medicinal parts of medicinal plants varies greatly with growth period. The secondary metabolites of plants are usually the active components in medicinal materials and are the material basis for the quality of medicinal materials. Therefore, the appropriate harvesting period based on the accumulation pattern of pharmacodynamic substances can effectively control and ensure the quality of medicinal materials [31]. In this study, six main flavonoids in *L. indica* L. cv. Mengzao at different harvest periods were determined by UPLC-QqQ-MS (Figure 5). It can be seen from Figure 5A and Appendix A that the content of flavonoids in the roots of *L. indica* L. cv. Mengzao showed a decrease-increase-decrease trend with delay of harvest time and reached its peak value at the vegetative stage, up to 179.74 µg/g. Appendix A showed that rutin was the main flavonoid in the roots, and the content of rutin was up to 172.09 µg/g at the vegetative stage. However, quercetin and luteolin were not detected during the whole growth period. In summary, we found that the contents of six flavonoids in the roots of *L. indica* L. cv. Mengzao reached the highest level during the vegetative growth period, and then showed a downward trend. The reason may be that with the increase of summer temperature, nutrients in roots accumulate rapidly to prepare for the germination of aboveground parts of plants. After that, due to the high temperature and strong light in July and August, aboveground vegetative growth and reproductive growth occurred at the same time, resulting in strong photosynthesis in the leaves [32]. Due to the organic combination of the aboveground part and the roots of the plant, during the vigorous growth of the aboveground part, the effective components of the roots of *L. indica* L. cv. Mengzao also reached the highest level in the year (vegetative growth period). Therefore, the suitable harvest time for flavonoids from the roots of *L. indica* L. cv. Mengzao should be in the vegetative stage.

The pattern of change of six flavonoids in the aboveground parts (i.e., stems, leaves, and flowers) of *L. indica* L. cv. Mengzao was inconsistent. Flavonoids in the stems and flowers reached a peak at initial flowering stage, while those in leaves reached the peak at the peak flowering stage. Specifically, in the whole growth period, the content of flavonoids in the stems of *L. indica* L. cv. Mengzao increased first and then decreased, reaching the peak value at the initial flowering stage, and up to 707.42 µg/g. The results are shown in Figure 5B and Appendix A.

In the whole growth period, the content of flavonoids in the leaves of *L. indica* L. cv. Mengzao showed a decrease-increase-decrease trend, reaching the highest at the peak flowering stage, with a content as high as 981 µg/g (Figure 5C). Rutin accounted for the main contribution, with the content reaching 782.66 µg/g. Table 5 shows that the highest contents of apigenin, kaempferol, luteolin, and luteolin-7-*O*-glucoside were mainly concentrated in the late growing season, which were 19.00 µg/g at the peak flowering stage, 9.31 µg/g at the filling stage, 122.49 µg/g at the peak flowering stage, and 56.55 µg/g at the filling stage, respectively.

Similarly, the main flavonoids in flowers are rutin, with content up to 1487.90 µg/g. Figure 5D shows that the content of six flavonoids in flowers reached the peak value at the initial flowering stage, with the content up to 2323.98 µg/g. The content of each part followed the order flowers > stems > leaves > roots, which may be due to the biosynthesis of flavonoids. For the stems, as a transmission organ, the nutrients in the flowers can be transferred to the roots, and the nutrients can also be synthesized and transmitted to each other in the aboveground parts and underground part [33], which suggests why sometimes the content for flavonoids in the stems is higher than that in the leaves. In general conditions, the photosynthesis and enzyme activity of plants have an impact on the growth and development of plants and the accumulation of effective ingredients [34,35]. In our study, we found that the main accumulation part of flavonoids in *L. indica* L. cv. Mengzao was the leaves. The reason for this is that, as the plant grows and develops, the leaves of *L. indica* L. cv. Mengzao gradually become larger, stomata open up, carbon dioxide uptake in the intercellular space increases, and photosynthetic rates increase. They can affect the synthesis of flavonoids. Although the leaves of *L. indica* L. cv. Mengzao become smaller, yellow, and old at the flowering stage, the carbon accumulated by plant photosynthesis can satisfy the synthesis of secondary metabolites that use carbon as a substrate [36], allowing the flavonoid content to remain increased. In addition, the biosynthesis of flavonoids requires the participation of many catalytic enzymes, and different enzymes have their own appropriate temperature for catalysis [37,38]. Under summer heat stress, the accumulation of some flavonoids gradually increased as plant phenylalaninammo-nialyase increased and peroxidase and polyphenol oxidase activities decreased [39,40]. In our study, the leaves were a rich accumulator of flavonoids, and the content reached its highest at the full flowering stage, which was also the optimal harvest period.

## 3. Materials and Methods

### 3.1. Plant Materials

The test material was the nationally approved variety (*Lactuca indica* L. cv. Mengzao), which was bred by Professor Zhang Xiufen of the former Inner Mongolia Institute of Agriculture and Animal Husbandry. *L. indica* L. cv. Mengzao was planted in the pasture base of Inner Mongolia Agricultural University on 10 May 2020, watered and weeded in time, and sampled from July to September. The samples were divided into roots, stems, leaves, and flowers, dried in the shade at room temperature, crushed, and stored for subsequent test and analysis. The features of the various parts of *L. indica* L. cv. Mengzao samples are shown in Figure 1. Information about these samples is summarized in Table 6.

### 3.2. Chemicals and Reagents

Methanol, acetonitrile, and formic acid (HPLC grade) were all purchased from Thermo Fisher Scientific (San Jose, CA, USA). Ethanol, ferric chloride, hydrogen peroxide, salicylic acid, and sodium carbonate were of analytical grade and were all purchased from Sinopharm Chemical Reagent Co., Ltd. (Shanghai, China). Chemical standards, including luteolin (lot B20888, purity >98%), rutin (lot B20771, purity >98%), quercetin (lot B20527, purity >98%), luteolin-7-*O*-glucoside (lot B20887, purity > 98%), apigenin (lot B20981, purity >98%), kaempferol (lot B21126, purity >98%), and 2-Chloro-l-phenylalanine (lot B25643, purity >98%), were all purchased from Yuanye Biotechnology Co., Ltd. (Shanghai, China).

### 3.3. Preparation of Samples Solution

The whole plant of *L. indica* L. cv. Mengzao and its different parts (roots, stems, leaves, and flowers) were weighed (1.0 g) and ultrasonically extracted (400 W) at room temperature for 30 min with 25 mL of 60% ethanol. A certain volume of 2-Chloro-l-phenylalanine (1 μg/mL) was added to the whole plant sample and centrifuged at 3000 rpm for 10 min and filtered through a 0.22-µm membrane filter. All of the sample solutions were stored at 4 °C before injection into the UPLC-MS/MS.

### 3.4. Chromatographic and Mass Spectrometry (MS) Conditions

#### 3.4.1. UPLC-Q-Exactive-MS/MS Method for Qualitative Analysis

The identification of flavonoids in *L. indica* L. cv. Mengzao was carried out using a Thermo UltiMate 3000 UPLC system coupled to a TSQ Quantum Access Max triple-stage quadropole mass spectrometer (Thermo Fisher Scientific, Waltham, MA, USA). A 20-μL aliquot of each sample solution was injected and analyzed on a Zorbax C_18_ column (100 × 2.1 mm, 1.8 μm, Agilent, Palo Alto, CA, USA). The separation was conducted at 30 °C (column temperature) using a gradient elution method with 0.1% formic acid in distilled water (solvent A) and acetonitrile (solvent B). The solvent gradient in volumetric ratios was set as follows: 0~2 min, 95% A; 2~6 min, 95~70% A; 6~7 min, 70% A; 7~12 min, 70~5% A; 12~13 min, 5% A; 13~16 min, 5~95% A; and 17 min, 95%. The flow rate was 0.3 mL/min, and the sample injection volume was 5 µL.

The electrospray ionization (ESI) and high-energy collisional dissociation (HCD) were applied in the positive ion mode and negative ion mode for the MS analysis. The operation conditions of mass analysis were set as follows: vaporizer temperature, 325 °C; sheath gas (N_2_) pressure, 45 arbitrary units; auxiliary gas (N_2_) pressure, 15 arbitrary units; spray voltage, 3.0 KV (PI)/3.2 KV (NI); capillary temperature, 350 °C; S-Lens RF Level: 55%; primary full scan (*m*/*z* 100~1500) and data-dependent secondary mass spectrometry (dd-ms2, Topn = 10); resolution, 120,000 (primary mass spectrometry) and 60,000 (secondary mass spectrometry).

#### 3.4.2. UPLC-QqQ-MS/MS Method for Quantitative Analysis

The UPLC-QqQ-MS/MS conditions were used for the determination of the flavonoids. An UltiMate 3000 system coupled to a TSQ Quantum Access Max triple-stage 308 quadropole mass spectrometer (Thermo Fisher Scientific, Waltham, MA, USA) was used. The separations were conducted at 25 °C on a Thermo-C_18_ column (100 × 2.1 mm, 1.8 μm, Waltham, MA, USA). The mobile phase was composed of 0.1% formic acid in distilled water (solvent A) and acetonitrile (solvent B). The solvent gradient in volumetric ratios was set as follows: 0~2 min, 95% A; 2~4 min, 95~70% A; 4~7 min, 70~60% A; 7~10 min, 60~20% A; 10~13 min, 20~5% A; 13~15 min, 5% A; 15~17 min, 5~50% A; and 17~20 min, 50~95% A. The flow rate was 0.3 mL/min, and the sample injection volume was 5 µL.

The mass analysis was adopted in positive and negative ionisation modes. The MS spectra were acquired in MRM mode. The MS conditions were optimised as follows: vaporizer temperature, 350 °C; capillary temperature, 300 °C; spray voltage, 3000/2500 V; shealth gas pressure, 35 Arb; and aux gas pressure, 10 Arb.

### 3.5. Validation of the Methods

#### 3.5.1. Linearity, Limit of Detection (LOD), and Limit of Quantification (LOQ)

The linearity was determined with six different concentrations of individual reference compounds. The six standard compounds were weighed and mixed, and the lowest concentration of the working solution for calibration use was gradually diluted with methanol to a series of concentrations. LODs and LOQs for each standard substance were acquired while the S/Ns were 3 and 10, respectively.

#### 3.5.2. Precision, Repeatability, Stability, and Recovery

The precision was analyzed using standard solutions with six replicates, and the RSD of the peak area of each standard compound was calculated. To confirm repeatability, six different sample solutions were prepared for the same sample (sample Y1, the *L. indica* L. cv. Mengzao, which was harvested at the vegetative stage) and the RSD of each component was calculated. According to the above chromatographic conditions, the sample Y1 solution was injected at 0, 2, 4, 8 and 12 h, respectively, and RSD was calculated to evaluate stability. A spike recovery test was carried out by adding the corresponding standard at low (80%), medium (100%), and high (120) levels to the raw materials (sample Y1 preparation), then measured in six duplicates. The spiked samples were extracted and determined according to the above method, and the spike recoveries were calculated using the following formula:Recovery % = [(measured amount − original amount)/amount added] × 100%

### 3.6. Sample Determination

All samples were extracted according to the method described in Section 3.3 and determined according to the chromatographic conditions in Section 3.4 to obtain the peak area. The corresponding concentration was calculated according to the regression equation of the standard curve.

### 3.7. Statistical Analysis

Data were statistically analyzed with SPSS software (version 13.0, Funduc software Inc., Livonia, MI, USA). The figure was drawn by OriginPro software (version 2021, Origin Lab®, Northampton, MA, USA). One-way analysis of variance (ANOVA) was carried out to calculate the statistical significance of obtained data. Data are expressed as the mean ± standard deviation of triplicate measurements.

## 4. Conclusions

Our method provides valuable information of the flavonoids in *L. indica* L. cv. Mengzao. In this study, 15 flavonoids of *L. indica* L. cv. Mengzao were identified and six main active ingredients were determinated by the UPLC-MS/MS technology. As far as the authors are aware, this is the first report on the distribution and dynamic changes of the chemical components in different parts of *L. indica* L. cv. Mengzao at different harvest periods. During the nutritional growth period, flavonoids, especially rutin, accumulated mainly in the roots. As the plant grows and develops, six species gradually accumulate towards the leaves and flowers. Our report provides an effective reference for the dynamic selection of the best harvest period based on the accumulation of the target ingredients. Flowers and leaves were the main harvesting parts of *L. indica* L. cv. Mengzao, and the flowering period was the most suitable harvesting period. In addition, the UPLC-MS/MS method developed in this study can be used for future research on the preparation of Ixeris polycephala Cass and its medicines.

## Figures and Tables

**Figure 1 molecules-26-07445-f001:**
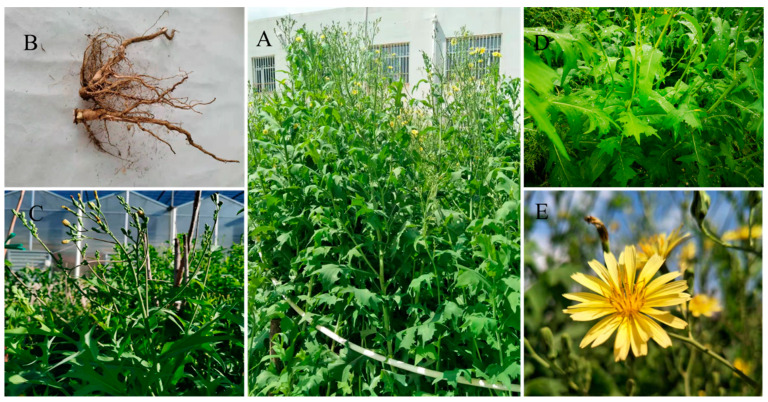
The various parts of *L. indica* L. cv. Mengzao samples. (**A**) Whole plants; (**B**) roots; (**C**) stems; (**D**) leaves; (**E**) flowers.

**Figure 2 molecules-26-07445-f002:**
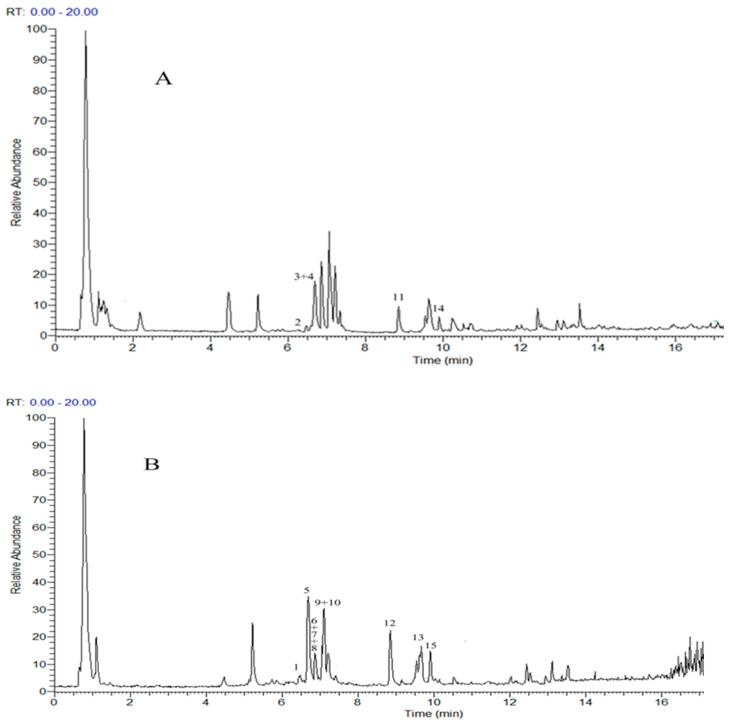
The flavonoids’ ion flow diagram of *L. indica* L. cv. Mengzao based on UPLC-Q-Exactive-Orbitrap-MS ((**A**): positive scanning mode; (**B**): negative scanning mode).

**Figure 3 molecules-26-07445-f003:**
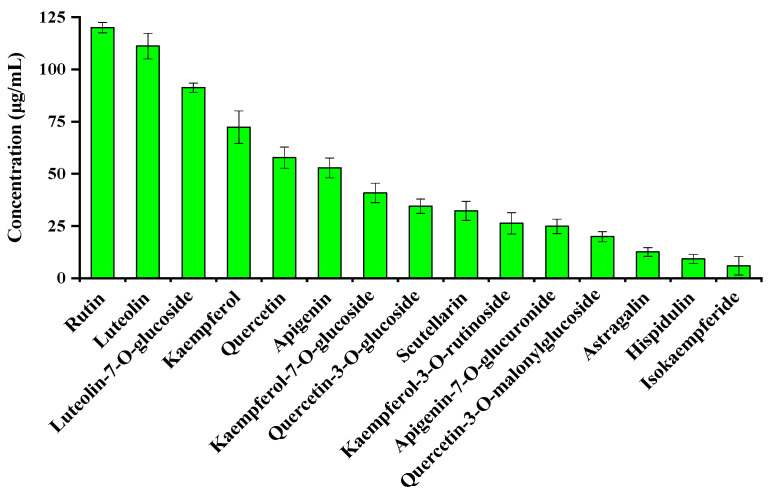
Concentration of 15 flavonoids in *L. indica* L. cv. Mengzao.

**Figure 4 molecules-26-07445-f004:**
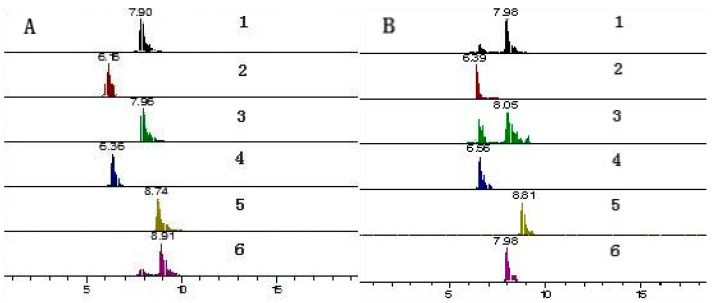
Multiple reaction monitoring (MRM) chromatograms of luteolin (1); rutin (2); quercetin (3); luteolin-7-*O*-glucoside (4); apigenin (5); kaempferol (6); (**A**) standard solution; and (**B**) *Lactuca indica* L. cv. Mengzao samples.

**Figure 5 molecules-26-07445-f005:**
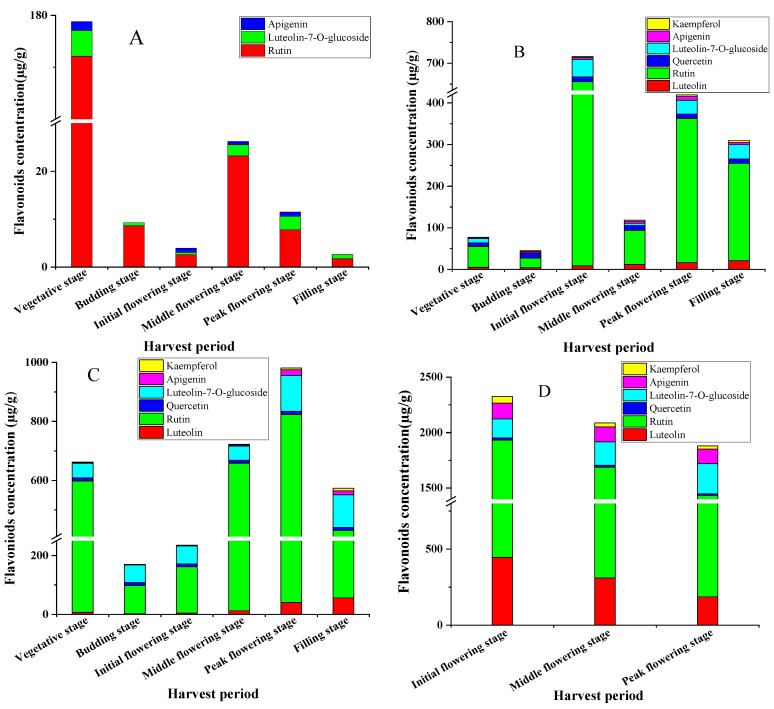
Dynamic accumulation of six flavonoids in roots (**A**), stems (**B**), leaves (**C**), and flowers (**D**) of *L. indica* L. cv. Mengzao.

**Table 1 molecules-26-07445-t001:** Characteristics of 15 flavonoid peaks from *L. indica* L. cv. Mengzao by UPLC-Q-Exactive-Orbitrap-MS.

No	tR/min	Molecular Formula	Ion mode	Mass (*m*/*z*) (∆ ppm)	MS/MS Fragment Ions (*m*/*z*)	Identification
1	6.468	C_27_H_30_O_16_	[M − H]^−^	609.14600 (−1)	301.03488, 300.02719, 210.46393, 178.99763, 151.00256	Rutin
2	6.473	C_27_H_30_O_15_	[M + H]^+^	595.16504 (−2)	433.10678, 287.05429	Kaempferol-3-*O*-rutinoside
3	6.682	C_21_H_20_O_12_	[M + H]^+^	463.08777 (−1)	301.03519, 271.02454, 178.99773, 151.00270	Quercetin-3-*O*-glucoside
4	6.690	C_21_H_20_O_11_	[M + H]^+^	449.10696 (−1)	287.05417, 210.24162, 153.01753	Luteolin-7-*O*-glucoside
5	6.694	C_21_H_20_O_11_	[M − H]^−^	449.10672 (−2)	287.05426, 153.0157, 116.17660, 88.12576	Kaempferol-7-*O*-glucoside
6	6.712	C_15_H_10_O_6_	[M − H]^−^	285.04037 (1)	268.68231, 151.50806, 133.02843	Luteolin
7	6.717	C_21_H_18_O_12_	[M − H]^−^	461.07242 (−1)	285.04025, 217.08017, 193.48659, 113.0233, 85.02839	Scutellarin
8	6.898	C_24_H_22_O_15_	[M − H]^−^	549.08826 (−2)	301.03476, 213.56686, 151.00270, 104.41696	Quercetin-3-*O*-malonylglucoside
9	7.061	C_21_H_20_O_11_	[M − H]^−^	447.09302 (−2)	285.04022, 255.02945, 151.00255	Astragalin
10	7.211	C_21_H_18_O_11_	[M − H]^−^	447.09149 (1)	271.05927, 153.01793, 113.02331, 85.02838	Apigenin 7-*O*-glucuronide
11	8.861	C_15_H_10_O_6_	[M + H]^+^	287.05423 (−1)	184.68994, 153.01776, 87.57372	Kaempferol
12	8.958	C_15_H_10_O_7_	[M − H]^−^	301.03506 (−1)	273.04022, 178.99767, 151.00264, 121.02842, 107.01268	Quercetin
13	9.682	C_15_H_10_O_5_	[M − H]^−^	269.04504 (−2)	151.00225, 107.01238	Apigenin
14	9.834	C_16_H_12_O_6_	[M + H]^+^	301.07016 (−1)	286.04651, 245.09196, 205.72273	Isokaempferide
15	9.838	C_16_H_12_O_6_	[M − H]^−^	299.05594 (−1)	284.03229, 256.03735, 215.19597, 108.85090, 72.84940	Hispidulin

**Table 2 molecules-26-07445-t002:** MS detection parameters of six compounds in *Lactuca indica* L. cv. Mengzao.

Flavonoids	t_R_ (min)	Molecular Formula	*m*/*z*	Quantitative Ion	Collision Energy (V)	Ion Mode
Luteolin	7.90	C_15_H_10_O_6_	287.10	153.05	30	ES^+^
Rutin	6.16	C_27_H_30_O_16_	609.30	299.89	36	ES^−^
Quercetin	7.96	C_15_H_10_O_7_	301.09	151.13	24	ES^−^
Luteolin-7-*O*-glucoside	6.36	C_21_H_20_O_11_	449.20	287.04	19	ES^+^
Apigenin	8.74	C_15_H_10_O_5_	271.09	149.06	26	ES^+^
Kaempferol	8.91	C_15_H_10_O_6_	285.07	187.02	31	ES^−^

**Table 3 molecules-26-07445-t003:** Calibration curves, LOD, and LOQ of six references.

Flavonoids	Calibration Curves	R	Linear Range /(µg/mL)	LOD /(ng/mL)	LOQ /(ng/mL)
1. Luteolin	Y = 719.907∗X + 117,248.00	0.9991	0.08~5.12	3.08	10.27
2. Rutin	Y = 139,827∗X + 10,171.60	0.9994	0.09~5.81	3.11	10.37
3. Quercetin	Y = 485.570∗X − 110,558.00	0.9990	0.21~1.10	10.32	34.4
4. Luteolin-7-*O*-glucoside	Y = 2473.043∗X + 123,989.00	0.9993	0.07~4.80	4.63	15.43
5. Apigenin	Y = 1306.807∗X − 12,189.90	0.9992	0.02~0.30	1.67	5.58
6. Kaempferol	Y = 215,145∗X − 2641.73	0.9992	0.05~0.65	2.08	6.93

**Table 4 molecules-26-07445-t004:** Precision, repeatability, stability, and recovery of six references.

Flavonoids	Precision RSD (%, *n* = 6)	Repeatability RSD (%, *n* = 6)	Stability RSD (%, *n* = 6)	Recovery (%, *n* = 3)	
				Mean	RSD
1. Luteolin	3.64	2.69	2.49	94.09	1.75
2. Rutin	4.08	1.93	4.56	105.35	1.52
3. Quercetin	2.15	2.32	1.74	99.41	2.13
4. Luteolin-7-*O*-glucoside	3.53	1.15	1.82	95.21	1.37
5. Apigenin	2.88	3.92	1.35	100.05	2.94
6. Kaempferol	1.85	3.08	2.47	101.85	1.46

**Table 5 molecules-26-07445-t005:** Dynamic accumulation of six compounds in leaves of *Lactuca indica* L. cv. Mengzao (µg/g).

Harvest Time	Luteolin	Rutin	Quercetin	Luteolin-7-*O*-glucoside	Apigenin	Kaempferol
Vegetative stage	7.344 ± 0.84 ^d^	590.869 ± 11.33 ^c^	10.324 ± 0.29 ^a^	49.729 ± 1.02 ^d^	2.404 ± 0.37 ^cd^	1.822 ± 0.14 ^d^
Budding stage	2.757 ± 0.52 ^f^	95.524 ± 5.20 ^f^	10.359 ± 0.08 ^a^	58.915 ± 2.21 ^c^	1.374 ± 0.03 ^d^	1.178 ± 0.09 ^e^
Initial flowering stage	4.494 ± 0.21 ^e^	157.022 ± 7.57 ^e^	10.288 ± 0.13 ^a^	60.413 ± 1.78 ^c^	1.640 ± 0.08 ^d^	1.323 ± 0.10 ^e^
Middle floweing stage	12.422 ± 0.01 ^c^	645.878 ± 19.55 ^b^	10.247 ± 0.16 ^a^	48.232 ± 1.19 ^d^	3.221 ± 0.03 ^c^	2.596 ± 0.19 ^c^
Peak flowering stage	40.602 ± 5.46 ^b^	782.662 ± 21.07 ^a^	10.348 ± 0.04 ^a^	122.487 ± 10.23 ^a^	18.999 ± 0.94 ^a^	6.053 ± 1.33 ^b^
Filling stage	56.552 ± 0.96 ^a^	374.052 ± 5.65 ^d^	10.401 ± 0.06 ^a^	109.959 ± 9.41 ^b^	13.594 ± 0.48 ^b^	9.308 ± 0.36 ^a^

Note: Different lowercase letters in the same column indicate the significance of the difference between treatments at the level of 0.05.

**Table 6 molecules-26-07445-t006:** Harvest time, growth period, and harvest parts of *Lactuca indica* L. cv. Mengzao.

Harvest Time	Phenology	Harvest Part
4 July 2020	Vegetative period	Roots, stems, leaves
14 July 2020	Budding stage	Roots, stems, leaves
25 July 2020	Initial flowering stage	Roots, stems, leaves, flowers
4 August 2020	Middle floweing stage	Roots, stems, leaves, flowers
14 August 2020	Peak flowering stage	Roots, stems, leaves, flowers
24 August 2020	Filling stage	Roots, stems, leaves

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
