# Peer review of "Comparative Analysis of Major Flavonoids among Parts of *Lactuca indica* during Different Growth Periods"

_molecules, 2021, doi:10.3390/molecules26247445_

Round 1

Reviewer 1 Report

Dear Editor

Hao et al investigate the flavonoids in  Lactuca indica L.cv. Mengzao distributed in its various parts at different 19 growth periods. Results indicated that fifteen flavonoids were identified from L. indica L.cv. Mengzao.

Finally, the submitted article  can be accepted after the major revision. 

I wonder why the six flavonoids some of them gave positive ion mode but the others were higher in negative ion mode in the mass spectrum.

  • Some fragments in MS/MS needed to be more discussed as its with RT are the real documents of the isolated compounds.
  • For table 1, as Rutin has short retention time but Hispidulin or Kaempferol came late.
  • Why not the researcher applies for Co-HPLC for the pure flavonoids which bought from the company with the extended one(extracted from the target plant).
  • The authors outline the main accumulation part for the Flavonoids the content in leaves reached its highest at the full flowering stage, which was also the optimum harvest time. The accumulation due to the biosynthesis and for the temperature in summer. I hope to outline how about the other factors, how about the other Climatic Factors like the percentage of Oxygen or Carbon dioxide other factor like enzymatic reasons.

There are some typing mistakes we discuss as fellow:

Line 3 Lactuca indica L.cv. Mengzao during Different Growth Periods.

Line 3 Lactuca indica during Different Growth Periods. As the title is too long.

Line 16 Abstract: Lactuca indica L.cv. Mengzao, a medicinal plant of   the Ixeris genus, is rich in

Line 17 flavonoids. In order to thoroughly analyze the flavonoids in L. indica L.cv. Mengzao, and

Line 18 the distribution and dynamic change of components in its various parts at different

Line 19 growth periods,

The sentence is too long and have a lots of and.

Line 19  The flavonoid extracts were identified and quantitatively

24 parts of L. indica L.cv. Mengzao followed the order flower> leaf> stem> root.

flowers> leaves> stems> roots.

26 is the most suitable harvesting period. This study provided provides valuable information for the

27 development and utilization of flavonoids in L. indica L.cv. Mengzao, and determined the

28 best part to harvest and the best time for harvesting.

The sentence is too long and have a lots of and.

33 Ixeris Cass. genus. It is a native to Europe and Central Asia, and is now mainly distributed

34 in China, Japan and Vietnam [1].

35 heat-clearing and detoxification, stopping pain, cool blood and anti-inflammation [add reference].

36 (200-300 A.D., Han Dynasty) the

39 thousands of years [add reference]. Modern Recently, phytochemical studies show that flavonoids, triterpenoids

43 cardiocerebral vascular system [4] and liver protection [5], effects that have aroused the (what does it mean?)

53 on the flavonoids of plants of the Ixeris genus.

56 opyranoside ethyl ester, 5,7-dihydroxy-4'-methoxy-flavone-7-O-rutinoside [2, 7-9].

58 Fla-

59 vonoids are complex and diverse, and are difficult to extract and determine. Thus, it is

60 necessary to establish a rapid and sensitive method for the determination and quantify

Sentence is too long (too much and).

67 contents of major flavonoids in its root, stem, leaf and flower at different growth stages

(roots, stems, leaves and flowers)

78  The results of

79 this study provide  (provides) a theoretical basis for carrying out diversified and deep processing of

80 forage (for age) and improving the added value of forage products, and may provide a reference

81 for quality monitoring of Ixeris polycephala products and further development and

82 utilization of Ixeris polycephala resources.

Sentence is too long (too much and).

84 Figure 1. The various parts features of L. indica L. cv. Mengzao samples. (A) whole plants; (B)

85 roots; (C) stems; (D) leaves; (E) flowers.

(Move to be on page 2 under the figure 1)

Summarize the sentence from line 106 until line 168 to be more organized and clearer. For example put the compounds which gave [M+H]+ together and [M-H]- in one group. Also, outline more fragments to be more satisfied about the identification of sample compound. Keep m/z in italic and the number of compounds in bold  

195 [27]. By comparing the separation effects of isocratic

195 [27]. Comparing the separation effects of isocratic

203 Kaempferol 8.91 C15H10O6 285.07 187.02 31 ES-

Keep the format of Table 2.

222 2.5.3 Dynamic Characteristics of Six Major Flavonoids in Different Parts of L. indica L.cv.

223 Mengzao

Move to be under the table

249 & 250 (stems, leaves and flowers)

255 give tab in the begging of the paragraph.

266 (flowers> stems> leaves>roots)

302 2-Chloro-L-Phenyalanine

302 2-Chloro-L-phenyalanine

318 N2 keep the style N2

344 All samples were extracted according to the method described in section 3.3, and deter

345 mined according to the chromatographic conditions in section 3.4 to obtain the peak area, and

352 Our new method provided identification information of the

Our newly technique provides valuable information of the…..

Reviewer 2 Report

The authors have studied the content of major flavonoids among parts of Lactuca indica L.cv. Mengzao during different growth periods. They indicate that phytochemicals from Ixeris polycephala and their related potential activities have not been fully explored therefore this study was done.

Minor comments

  1. Introduction line 34-35 sentence “It is widely used in folk medicine because of its functions of heat-clearing and detoxification, stopping pain, cool blood and anti-inflammation” is not in scientific language, if it is meant as a citation taken directly from folk medicine, then it should be indicated.
  2. Line 65- sentence is misleading because Fig. 1 shows pictures of the plant but not the determined flavonoids.
  3. In many cases, redundant words or whole parts of sentences are visible. As one example is the sentence in Line 101 ‘A total of 15 flavonoid peaks were detected identified in indica L.cv. Mengzao”.

Major

In Abstract Line 27 mentioned goal “This study provided valuable information for the

development and utilization of flavonoids in L. indica L.cv. Mengzao” is not reached as no information and study on utilization of the flavonoids.

Fig. 2. shows concentration which is not content as stands in the capture. Content should be expressed as amount per extract or wet plant weight.

Fig.4 what means harvest time expressed as decimal number?

Line 258-259. To my mind, Table S3 would be better to put in the article not in the supplement as “Table S3 shows that the contents of apigenin,

259 kaempferol, luteolin and luteolin-7-O-glucoside showed an overall upward trend…”

Line 350. What is statistically analysed and where statistical significance is shown in figures or tables?

“The data were statistically analysed by OriginPro 2021 (Origin Lab®, MA, USA)”.

Conclusion Line 353. What confirms statement about new method?” Our new method provided identification information of the flavonoids in L. indica

L.cv. Mengzao.”   UPLC-MS/MS method is not a new method.

The conclusions could be more specific, and it is desirable to include few numbers for key measurements. For example in the sentence, “The data not only showed that there were differences in the active ingredients of the aboveground and underground parts of L. indica L.cv. Mengzao ” how big were differences?

What was so specific and novel in flavonoid analysis “In addition, the UPLC-MS/MS method established”?

Round 2

Reviewer 2 Report

The manuscript is improved. I agree with made changes in the text and do not have comments.